# Utilizing Iron Ore Tailing as Cementitious Material for Eco-Friendly Design of Ultra-High Performance Concrete (UHPC)

**DOI:** 10.3390/ma14081829

**Published:** 2021-04-07

**Authors:** Gang Ling, Zhonghe Shui, Xu Gao, Tao Sun, Rui Yu, Xiaosheng Li

**Affiliations:** 1School of Materials Science and Engineering, Wuhan University of Technology, Wuhan 430070, China; linggang0609@163.com (G.L.); li-xiaosheng@whut.edu.cn (X.L.); 2State Key Laboratory of Silicate Materials for Architectures, Wuhan University of Technology, Wuhan 430070, China; zhshui@whut.edu.cn (Z.S.); sunt@whut.edu.cn (T.S.); r.yu@whut.edu.cn (R.Y.); 3Wuhan University of Technology Advanced Engineering Technology Research Institute of Zhongshan City, Xiangxing Road 6, Zhongshan 528400, China; 4School of Civil Engineering and Architecture, Wuhan University of Technology, Wuhan 430070, China

**Keywords:** iron ore tailing (IOT), cementitious materials, ultra-high performance concrete (UHPC), eco-friendly

## Abstract

In this research, iron ore tailing (IOT) is utilized as the cementitious material to develop an eco-friendly ultra-high performance concrete (UHPC). The UHPC mix is obtained according to the modified Andreasen and Andersen (MAA) packing model, and the applied dosage of IOT is 10%, 20%, and 30% (by weight), respectively. The calculated packing density of different mixtures is consistent with each other. Afterwards, the fresh and hardened performance of UHPC mixtures with IOT are evaluated. The results demonstrate that the workability of designed UHPC mixtures is increased with the incorporation of IOT. The heat flow at an early age of designed UHPC with IOT is attenuated, the compressive strength and auto shrinkage at an early age are consequently reduced. The addition of IOT promotes the development of long-term compressive strength and optimization of the pore structure, thus the durability of designed UHPC is still guaranteed. In addition, the ecological estimate results show that the utilization of IOT for the UHPC design can reduce the carbon emission significantly.

## 1. Introduction

Ultra-high performance concrete (UHPC) is a newly developed construction material, which is characterized by superior properties [1,2,3]. Owing to its remarkable performances, the service life could be prolonged and the relieve maintenance costs must be reduced significantly when UHPC is utilized in the construction of concrete structure and engineering [4]. Hence, the research and application in engineering of UHPC is becoming more and more extensive. A typical UHPC mixture comprises of cement, fly ash (FA), silica fume (SF), steel fiber, fine aggregates, superplasticizer, and water [5,6]. Generally, the production of UHPC consumes a large amount of energy and emits an abundance of CO_2_, since the UHPC mix contains huge amounts of cementitious materials (800–1000 kg/m^3^) [7,8].

Therefore, UHPC is still treated as a kind of building material which does not conform to the concept of sustainable development [9,10]. In addition, the hydrated cement in UHPC is only accounting for 30–40%, which means that a substantial part of cements just plays the role of fillers in UHPC [11,12]. Consequently, it is logic to utilize some inactive fillers to partly replace the cement, which will reduce the environment burden and widen the application of UHPC [13].

Cement production is a traditional industry with high emission and high energy consumption, which has an important responsibility for the global greenhouse effect and energy crisis [14,15,16]. There is an available way to cut down the CO_2_ emission and energy consumption by replacing cement with suitable powders [17]. In the preparation of concrete, supplementary cementitious materials (SCMs) are used to replace cement. For instance, silica fume (SF) [18], fly ash (FA) [19,20], ground granulated blast slag (GGBS) [15,16], and husk ash [21,22,23]. The literatures indicate that a moderate dosage of SCMs in concrete can not only reduce the amount of cement, but also optimize its workability, mechanical properties, volumetric stability, and durability [19,24,25,26]. With the extensive application of SCMs in the concrete industry, a great deal of industrial by-products is rapidly consumed. As a result, the quality and stability of SCMs has gradually become a concern [27,28]. For instance, high-grade fly ash has been a scarce material in China. In addition, SF is an important component of high-performance concrete but its price is also very expensive. Hence, it is necessary to seek more available and cheaper SCMs in the preparation of UHPC [29,30].

Iron ore tailing (IOT) is the mineral wastage which comes from the beneficiation process of iron ore [31,32,33]. The discharge amount of iron ore tailings is about 600 million tons per year in China [34]. However, the main treatment method of iron ore tailing is stacking by natural accumulation as few effective utilization measures has been employed [32,33,35]. Thus, it causes serious waste of land resources, environmental pollution, and even endangers human health [32,33]. Therefore, the rational application of tailing is a significant subject with economic and environmental benefits. The chemical components of iron ore tailings are similar to the cement and other SCMs [11] and the physical characteristics are similar to natural sand [36]. Hence, iron ore tailings may be utilized in the preparation of building materials and is feasible. 

It has been demonstrated that iron ore tailings can be applied to produce concrete materials [11,31]. It is found that the mechanical property of engineered cementitious composites (ECC) containing iron ore tailings is comparable to the typical ECC with micro-silica sand [37]. The iron ore tailings have been applied as fine aggregates during the UHPC production, and the mechanical property of the designed UHPC is comparable to the control UHPC when the substitution of the natural sand is no more than 40% [38]. In addition to being used as aggregates, the iron ore tailings can also be used as cementitious materials [39]. It is found that employing iron ore tailings to replace cement can optimize the mechanical properties of ordinary concrete [11,31,40], and the feasibility of adding IOT into ordinary concrete is pointed out [40]. However, there are significant differences in the design method and the composition of cementitious materials between UHPC and ordinary concrete. Therefore, the feasibility of preparing UHPC by utilizing a high volume of IOT to replace cement still needs demonstration. However, there is very limited research about utilizing iron ore tailings as cement replacement to prepare UHPC. According to the sustainable development concept of UHPC preparation with industrial waste, it is logical to explore the feasibility of adopting iron ore tailings rather than cement to prepare UHPC. In addition, there is a considerable part of un-hydrated cement in the UHPC mixture [12]. Thus, the application of iron ore tailings in UHPC is a prospective measure to treat them and improve the environmental friendliness of UHPC. In this way, significant economic and environmental benefits can be obtained.

According to the opinion mentioned above, iron ore tailings are incorporated to replace cement in the designed UHPC, the replacement is 10%, 20%, and 30% by weight, respectively. In addition, the influence of iron ore tailings on the designed UHPC is evaluated, including workability, rheology behavior, compressive strength, hydration process, pore size distribution, and chloride resistance. Moreover, the ecological effect of the designed UHPC with iron ore tailings is also appraised.

## 2. Materials and Methods

### 2.1. Materials

Portland cement (Huaxin, Huangshi, China) (P II 525 according to GB/T 175-2007) [41], SF (Southeast Star Technology Development Co., Ltd., Chengdu, China), and FA (Yangluo power plant, Wuhan, China) are utilized as cementitious materials in this study. The ground iron ore tailing (IOT) (Daye iron mine, Huangshi, China) is employed to replace the cement. The SEM images of the utilized IOT and cement are presented in Figure 1 and the phase identification by the XRD test of the exhibited IOT is shown in Figure 2. The chemical compositions of cementitious materials are shown in Table 1. The polycarboxylate superplasticizer (Sp) with a 20% solid content is used in the designed UHPC, and its water-reducing rate is more than 30%. Sieved natural river sand is used as a fine aggregate.

### 2.2. Experimental Methodology

#### 2.2.1. Mix Design Method

The modified Andreasen and Andersen (MAA) model is utilized for the UHPC mix design in this study, and the detailed design steps are illustrated in the literature [6,42,43]. The designed mix proportion of UHPC is shown in Table 2. The optimized grading curves of each mixture (IOT0, IOT10, IOT20, and IOT30) are presented in Figure 3.

#### 2.2.2. Workability

Fresh UHPC is prepared according to the manufacturing procedure in the reference [44]. Then, the workability of fresh UHPC is evaluated according to BS-EN-1015-3 [45]. The matrix is lifted perpendicularly after the fresh UHPC paste is poured into it, and then the diameter is measured when UHPC stops flowing. Measurements are taken twice along the vertical direction with the average value as the test result.

#### 2.2.3. Rheology Behavior

The R/S-SST soft-solid rheometer rotor viscometer (Brookfield Company, Toronto, ON, Canada) is adopted to measure the rheological property of fresh paste, and a 20 mm × 10 mm rotor is selected for testing. The range of shear rate is 0.01–1000 rpm and shear stress is 0–3.5 × 10^4^ Pa, respectively. The test procedure of the rheometer is presented in Figure 4.

#### 2.2.4. Mechanical Properties

The samples with the measurement of 40 mm× 40 mm× 160 mm are used for the compressive strength according to BS-EN-196-1 [46]. The samples are demolded after casting for 24 h, and then cured in the standard condition. The compressive strength of UHPC samples is tested after curing to a specific age. In this experiment, at least six specimens are tested for each group.

#### 2.2.5. Hydration Kinetics

The isothermal heat of each mixture is measured by the ATM AIR isothermal calorimeter. The specimens are prepared and shift to the containers, and then the containers are placed in the calorimeter. The isothermal heat of each sample is determined for 72 h.

#### 2.2.6. Pore Size Distribution

The mercury intrusion porosimetry (MIP) test is utilized to evaluate the pore size distribution of designed UHPC samples. The samples are cut into less than 5 mm fragments after curing for 28 ds, and then put in ethanol to stop their hydration. The specimens are dried for 24 h at 60 °C before testing. The Micromeritics AutoPore-9500 mercury intrusion porometer (Micromeritice, Norcross, GA, USA) is used for the MIP test, and its pressures range from 0.0034 to 379.225 MPa.

#### 2.2.7. Chloride Resistance

The chloride resistance of designed UHPC is evaluated by the coulomb electric flux test according to ASTM C1202 [47]. The cylindrical specimens with the measurement of Ф100 mm × 50 mm is used for the test after curing for 28 ds.

#### 2.2.8. Environmental Evaluation

The ecological effect of designed UHPC with IOT is evaluated by the calculation of the carbon foot print. The CO_2_ emissions of each raw material are according to reference [13]. The carbon foot print of designed UHPC is calculated by the accumulation of CO_2_ emission for the used material per cubic meter of UHPC.

## 3. Results and Discussion

### 3.1. Workability of Designed UHPC with IOT

Figure 5 demonstrates the workability of designed UHPC contents with different amounts of IOT. The results show that the workability of designed UHPC is improved with the incorporation of IOT. Compared with the slump flow value of the mixture without IOT (IOT0), the slump flow is improved by 7.7%, 19.2%, and 53.8%, when the dosage of IOT is 10%, 20%, and 30%, respectively. The results can be clarified as follows: The particles of the adopted IOT are larger than the cement, which correspond to the smaller specific surface area and lower water requirement. In addition, the IOT has a crystalline structure and lower water absorption [37,48]. Therefore, the water requirement of cementitious materials is decreased and more free water is released. As a result, the workability of designed UHPC is improved.

### 3.2. Rheology Behavior of Designed UHPC with IOT

The rheological behavior of designed UHPC with IOT is evaluated and exhibited in Figure 6. It is clear from Figure 6a that the viscosity of all UHPC mixtures decreased significantly with the increase of shear rate, which indicates that the rheological behavior of designed UHPC fits the characteristics of typical pseudoplastic fluids. The Herschel-Bulkley model [49] is utilized for the analysis of the rheological behavior, as shown in formula (1) and Figure 7.
(1)$${\tau =\tau }_{0}{+m\gamma }^{n}$$
where $\tau_{0}$ means the yield stress (Pa), τ means the shear stress (Pa), *m* means the consistency index (Pa·sn), means the shear rate (rad/s), n means the rheological behavior index. The plastic viscosity (μ) can be calculated by formula (2) [50].
(2)$$\mu =\frac{3m}{n+2}\gamma_{\max}^{n-1}\;$$
where γ_max_ means the maximum shear rate during testing. The measured and fitted rheological curves are shown in Figure 6b, it is clear that the correlation coefficients are over 0.99. The rheological parameters are calculated and displayed in Table 3. It can be found that the yield stress is decreased with the addition of IOT. For example, the yield stress of the UHPC mixture without IOT (IOT0) is 30.00 Pa. In addition, the yield stress decreases by 7.03%, 19.26%, and 36.87%, respectively, when the dosage of IOT increases from 10% to 30%. Hence, the workability of UHPC mixtures containing IOT is improved. This is owning to the increase of free water when the cement is replaced by IOT. The spacing of particles in the UHPC mixture is broadened accordingly, and the resistance to the paste flow is weakened [51]. In addition, Table 3 demonstrates that the plastic viscosity value is reduced when IOT is utilized in UHPC. However, the minimum plastic viscosity value is obtained when the addition of IOT is 10%. This is due to the fact that the fine IOT particles can fill in the interstice between the cementitious particles of the UHPC mixture. Thus, the packing density of the UHPC mixture is enhanced and the released free water is increased. It is important to notice that the lower plastic viscosity is conducive to a better filling capacity of UHPC [52].

Hence, it could be summarized from the slump flow and rheology tests that the addition of IOT is beneficial to reduce the viscosity and optimize the workability of UHPC.

### 3.3. Mechanical Properties of Designed UHPC with IOT

The compressive strength of designed UHPC with IOT at the age of 3, 7, and 28 d are tested and revealed in Figure 7. The results show that the compressive strength decreases gradually with the addition of IOT at an early age. For instance, when the substitution rate of cement replaced by IOT is 10%, 20%, and 30%, the compressive strength of 3 ds is reduced by 5.3%, 11.1%, and 21.7%, respectively. Referring to the compressive strength of 7 d, the value of the UHPC sample that contains 10% IOT is almost the same as the mixture without IOT (about 96.8 MPa). Then, when the addition of IOT is 20% and 30%, the compressive strength at 7 d decreases to 84.2 and 76.5 MPa, respectively. However, the 28 ds of compressive strength for IOT10 and IOT20 are 141.5 and 123.4 MPa, which are 16.3% and 1.2% higher than the IOT0 mixture, respectively. When the addition of IOT is by 30%, the compressive strength drops to 115.7 MPa, but it is still comparable to IOT0.

The IOT is a kind of supplementary cementitious material with low pozzolanic reactivity, leading to the reduction of generated hydration products at an early age (3 d) when the cement is displaced by IOT [11]. Therefore, the compressive strength of designed UHPC with IOT is decreased, and the reduction range of 3 d compressive strength becomes more significant with an increase of the IOT dosage. Nevertheless, the increase of compressive strength from 7 to 28 d for IOT0, IOT10, IOT20, and IOT30 are 25.7%, 45.1%, 46.5%, and 55.2%, respectively. The increment of UHPC with IOT is more significant. Based on the above experimental phenomenon, it is suggested that the improvements are due to the filling effect and pozzolanic effect of IOT [11,40,48]. These are beneficial for the compressive strength development of the hardened UHPC mixture. Moreover, the IOT has an appearance of an angular structure (as shown in Figure 1), so the adhesion between IOT and paste is enhanced. Consequently, the mechanical performance of designed UHPC samples is increased at the age of 28 d when 10% of the cement is replaced by IOT. When the addition of IOT is 30%, the packing density of UHPC mixtures will be affected since the particles size of IOT is larger than the cement. In addition, the dilution effect of IOT reduces the amount of generated hydration products of the designed UHPC mixture. These factors result in the depression of compressive strength for IOT30 at 28 d. The compressive strength tests are carried out on specimens of dimensions common to concrete in consideration of the size effect [53]. The measured compressive strength of UHPC samples is decreased by 10%~20%, but it is still higher than 100 MPa.

In summary, the compressive strength of designed UHPC at an early age is reduced with the addition of IOT, while the long-term performance of designed UHPC can be optimized when an appropriate amount (no more than 20%) of IOT is utilized. Hence, IOT is a king of suitable power material for the UHPC preparation.

### 3.4. Autogenous Shrinkage of Designed UHPC with IOT

The autogenous shrinkage of each mixture with a different dosage of IOT is measured to evaluate the volume stability of designed UHPC. The results are illustrated in Figure 8. It can be noticed that the development of autogenous contraction consists of three stages. That is, rapid growth stage, rebound stage, and gradual growth stage. These three processes are the reflex of a series of chemical and physical reactions in the UHPC system [54]. As commonly known, the rapid growth stage is crucial to the overall autogenous shrinkage of UHPC [5]. Accounting to fact that the autogenous shrinkage at an early age mainly comes from the tensile strain capacity [55,56], cracking easily occurs for UHPC at an early age. The results in Figure 8 indicate that the amount of autogenous shrinkage at the rapid growth stage is limited with the addition of IOT. 

The gradual growth stage of the self-deformation reflects the continuous evolution process of UHPC at a later age. It is clear that the curves of different mixtures of designed UHPC will tend to be parallel, and the contraction rate becomes similar accordingly. The autogenous shrinkage of designed UHPC at an early age is reduced by 22.3%, 29.5%, and 44.6% when the dosage of IOT is 10%, 20%, and 30% by comparing with the reference mixture. This should be attributed to the mechanisms as follows: (1) The content of cement is reduced and the hydration process at an early age is decreased when the cement is partly replaced by IOT, resulting in the depression of autogenous shrinkage. (2) According to Section 3.1, the effective water to cement ratio is increased when IOT is utilized to replace the cement. In addition, the content of capillary water w/c is raised. The evolution process of autogenous shrinkage is affected by the reduction of the water content [57,58]. Hence, the autogenous shrinkage of designed UHPC is moderated. Therefore, the experimental results indicate that utilizing IOT is effective in improving the volume stability of UHPC.

### 3.5. Isothermal Calorimetry of Designed UHPC with IOT

Figure 9 exhibits the effect of IOT on hydration heat for the designed UHPC, and the heat flow and cumulative heat are evaluated. It can be noticed that the maximum peak of heat flow for each designed UHPC mixture appears in the first 3 d (as shown in Figure 9a). That is, the main hydration of mixture occurs during this period [44]. In addition, it should be noticed that the maximum peak value of heat flow is decreased with an increase of IOT dosage. It can be noticed from Figure 9b that the effect of IOT on the cumulative heat is the same as that of the maximum peak value. This should be attributed to the fact that the activity of IOT is lower. The overall hydration reaction rate and the amount of generated hydration products are reduced, owning to the dilution effect of IOT. Then, the maximum peak value and cumulative heat of the samples with IOT is depressed [59].

The detailed isothermal calorimetry results of designed UHPC with IOT are listed in Table 4. The Q_min_ and Q_max_ in Table 4 mean the minimum and maximum heat release rate respectively, while the t(Q_min_) and t(Q_max_) in Table 4 represent the time when Q_min_ and Q_max_ are achieved, respectively. In particular, the specific time of t(Q_min_) is considered as the end of the induction period. The results in Table 4 indicate that the addition of IOT prolonged the induction period, but the Q_min_ is increased when the addition of IOT is 10% (IOT10). In addition, it can be observed that the t(Q_max_) is slightly advanced when the dosage of IOT is 10%. Then, the t(Q_max_) is gradually postponed with the addition of IOT. The heat flow of UHPC mixture with IOT is higher than the reference mixture (IOT0). Therefore, the later strength development of the UHPC samples with IOT is enhanced. These results could be explained as follows: (1) The nucleation effect of added IOT can accelerate the hydration reaction rate at an early age, which results in the increase of Q_min_ and the advance of t(Q_max_) for IOT10 [11]; (2) adding IOT will reduce the content of cement components, while increasing the available water for cement hydration. Thus, the needed time to reach the supersaturation critical value of calcium hydroxide precipitation is delayed [25]. The released free water can modify the hydration environment of cement and the hydration degree of cement is increased. Thus, the time to reach t(Q_max_) for IOT20 and IOT30 is delayed.

### 3.6. Pore Structure of Designed UHPC with IOT

Figure 10 illustrates the effect of IOT on the pore size distribution designed UHPC at 28 d. The results in Figure 10 show that the pore content of designed UHPC at 28 d is increased gradually with the addition of IOT. For instance, when the dosage of IOT is 10%, 20%, and 30%, the cumulative pore volume is increased by 2.8%, 7.7%, and 24.9% (compared with IOT0), respectively. However, it should be noticed that pores larger than 20 nm are reduced by 11.9–9.8% and pores smaller than 20 nm are increased by 19.7–36.3% when 10–20% of IOT is added into the mixture. Hence, the compressive strength of UHPC is increased when no more than 20% of IOT is added (as shown in Figure 7). When 30% of the cement is replaced by IOT, both the content of harmless pore (<20 nm) and less harm pore (20–50 nm) are significantly increased, while the volume of more harm pore (>50 nm) has a limited influence [60]. Thus, the compressive strength is comparable to the reference mixture (as shown in Figure 7). These results are related to the filling and nucleation effect of fine IOT practices [48]. In addition, the chemical activity (pozzolanic reaction) of IOT also has some beneficial effects [11,40,48]. However, the generated hydration products are insufficient when excessive cement is replaced by IOT [37]. Thus, the porosity of designed UHPC is increased. Hence, in order to obtain an appropriate density and microstructure, it is necessary to regulate the dosage of IOT.

### 3.7. Chloride Resistance of Designed UHPC with IOT

The effect of IOT on the chloride resistance coulomb of designed UHPC is evaluated by the coulomb electric flux test, which is adopted for the chloride permeability rating and the results are shown in Figure 11. It is clear that the designed HPC mixtures show excellent chloride resistance when the addition of IOT is no more than 20%. However, the chloride resistance of designed HPC mixtures is significantly decreased when the dosage of IOT is 30%. It can be found from Figure 11 that the total charge of IOT10 and IOT20 mixtures are 85 and 92 C, which is 4.1% and 13.6% higher than the reference mixture (81 C). A further addition of IOT (IOT30) leads to the increase of total charge to 107 C, and the increase is 32.1% compared with IOT0. It should be attributed to the low activity of IOT, leading to the decrease generated by the hydration products and deterioration of the microstructure of the designed UHPC. As a result, the durability of UHPC is reduced. Nevertheless, it should be noticed that the chloride permeability rating for almost all the designed UHPC containing IOT can be rated as “negligible”, and has great advantages when compared with the normal strength concrete. Consequently, IOT can be utilized in UHPC in the view of chloride ion-penetration resistance.

### 3.8. Environmental Evaluation of Designed UHPC with IOT

Compared with traditional concrete, the UHPC can be treated as eco-friendly materials owning to its outstanding mechanical properties. Thus, a lightweight can be realized and the volume of raw materials can be reduced during the construction of concrete structure. Moreover, its excellent durability can significantly prolong the service life and reduce the cost of later maintenance of concrete.

In addition to advantages mentioned above, the emission CO_2_ index of the designed UHPC with IOT needs to be evaluated as the cement is partially replaced by IOT. The carbon foot print is adopted to appraise the sustainability of designed UHPC, and the calculated emission of CO_2_ (kg/m^3^) is shown in Figure 12. It can be noticed that the addition of IOT cuts down the emission of CO_2_. For instance, the emission of CO_2_ for a typical UHPC mixture without IOT is 690.96 kg/m^3^ [61], this value is significantly reduced to 470.42 kg when the addition of IOT is 30%. In addition, the IOT has been utilized as the fine aggregate to replace the river sand. However, the emission of CO_2_ is still 672.06 kg/m^3^ when the replacement is 100% according to reference [61], since the CO_2_ emission for the UHPC mixture mainly comes from the manufacture of cement. Moreover, the utilization of IOT in UHPC can help relieve the landfills caused by IOT. In general, IOT can be used to prepare an eco-friendly UHPC with advanced properties. Hence, it is logical to use IOT as the cementitious materials from the perspective of an eco-friendly UHPC design.

## 4. Conclusions

The practicability of incorporating IOT as cementitious materials to prepare an eco-friendly ultra-high performance concrete (UHPC) has been evaluated in this research. According to the obtained results in this research, the main conclusions can be summarized:(1)The utilization of IOT optimizes the workability of fresh UHPC mixture by reducing its plastic viscosity and yield stress, which is mainly attributed to the larger particle size of IOT and increase of the released free water.(2)The addition of IOT reduces the compressive strength of designed UHPC at 3 d, but the compressive strength at 28 d is still comparable to the control mixture (IOT0) when the replacement of IOT is 30%. This is due to the fact that a low activity of IOT will weaken hydration at an early age, while the filling and nucleation effect of fine IOT particles will optimize the pore structure and improve its compactness at a later age. Thus, the development of the compressive strength for the designed UHPC with IOT is promoted.(3)The incorporation of IOT can reduce the autogenous shrinkage of designed UHPC significantly. This should be owning to the decrease of the cement content when IOT is added. Thus, the heat release rate at an early age is reduced. In addition, the released free water could postpone the reduction of internal relative humidity of UHPC. Hence, the utilization of IOT is beneficial to reduce the autogenous shrinkage of UHPC.(4)Based on the environmental evaluation of designed UHPC with IOT, the emission of CO_2_ of UHPC is reduced by 31.92% when 30% of the cement is replaced by IOT. In addition, the utilization of IOT in UHPC is beneficial to relieve the landfill problem caused by IOT. In terms of economic and environmental benefits, it can be summarized that the IOT is suitable to be used as the cementitious materials to design an eco-friendly UHPC with advanced properties.

## Figures and Tables

**Figure 1 materials-14-01829-f001:**
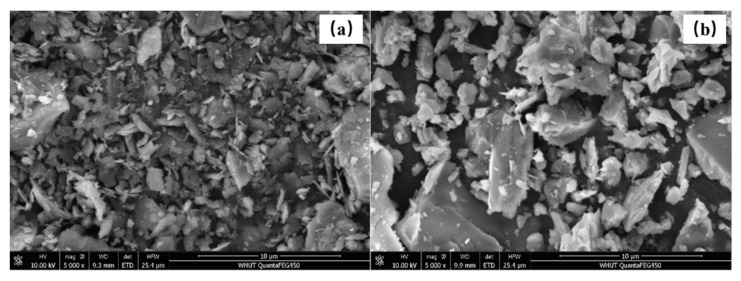
The SEM images of particle morphology for (**a**) iron ore tailing (IOT) and (**b**) cement.

**Figure 2 materials-14-01829-f002:**
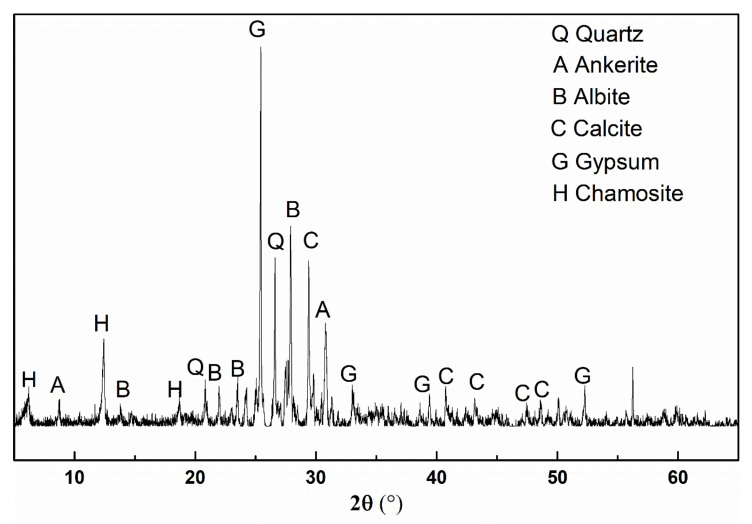
X-ray diffraction analysis of iron ore tailing.

**Figure 3 materials-14-01829-f003:**
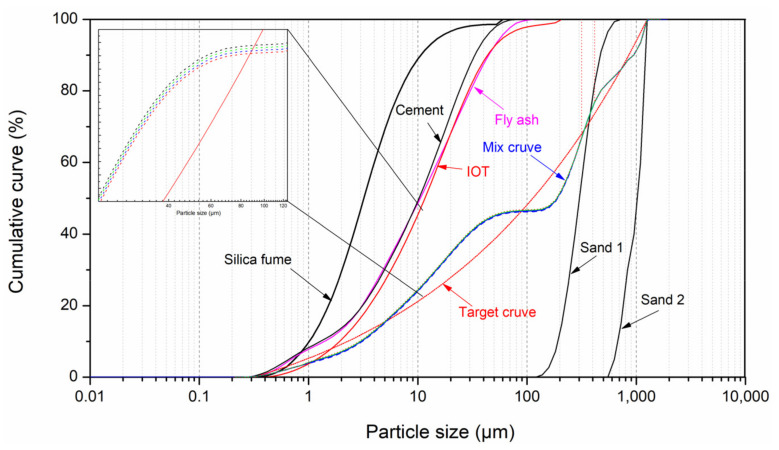
Particle size distributions of the ingredients, target, and optimized grading curves of different ultra-high performance concrete (UHPC) mixtures.

**Figure 4 materials-14-01829-f004:**
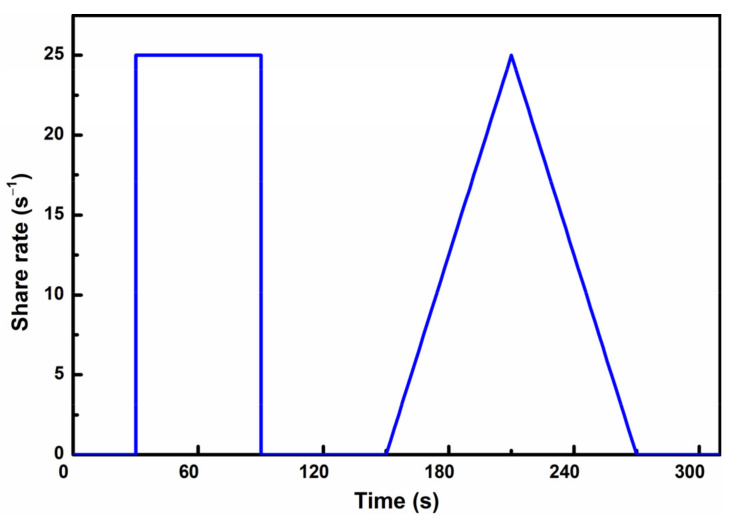
Test procedure for rheological properties.

**Figure 5 materials-14-01829-f005:**
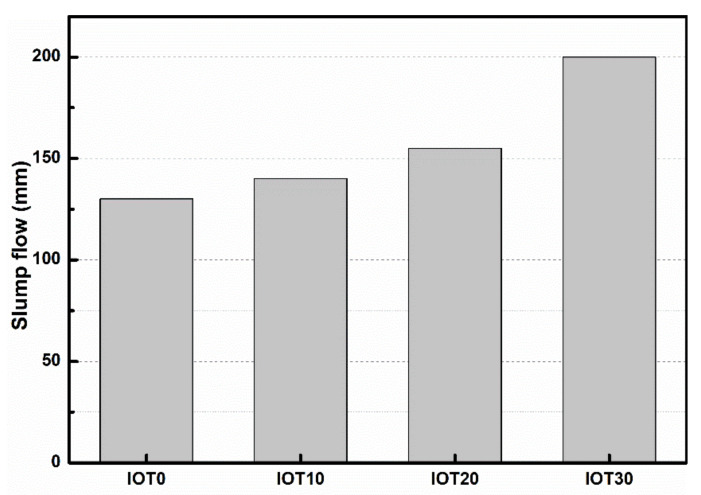
Effect of iron ore tailing on the slump flow of designed UHPC.

**Figure 6 materials-14-01829-f006:**
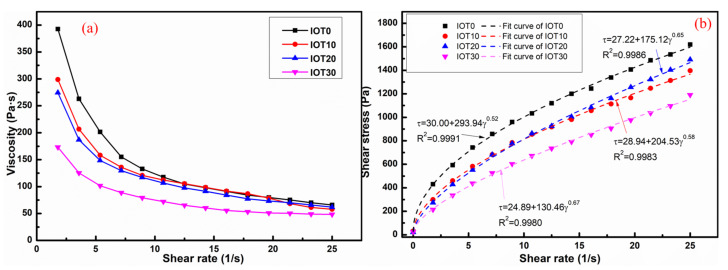
Effect of iron ore tailing on the rheology behavior of designed UHPC: (**a**) Viscosity; (**b**) shear stress.

**Figure 7 materials-14-01829-f007:**
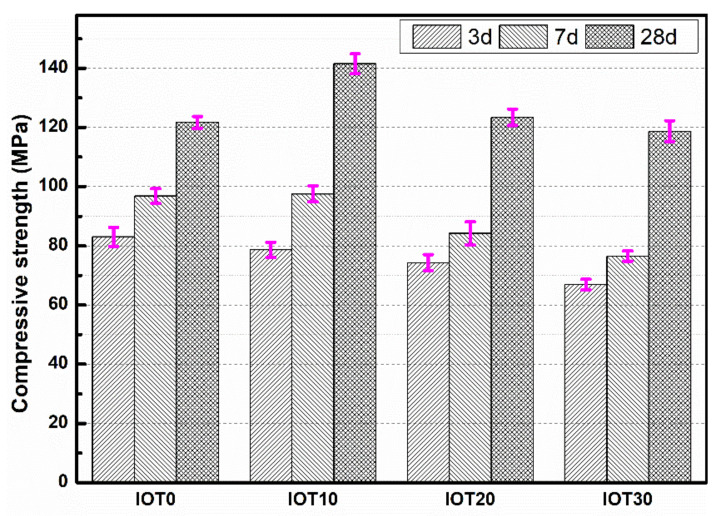
Effect of iron ore tailing on the compressive strength of the designed UHPC.

**Figure 8 materials-14-01829-f008:**
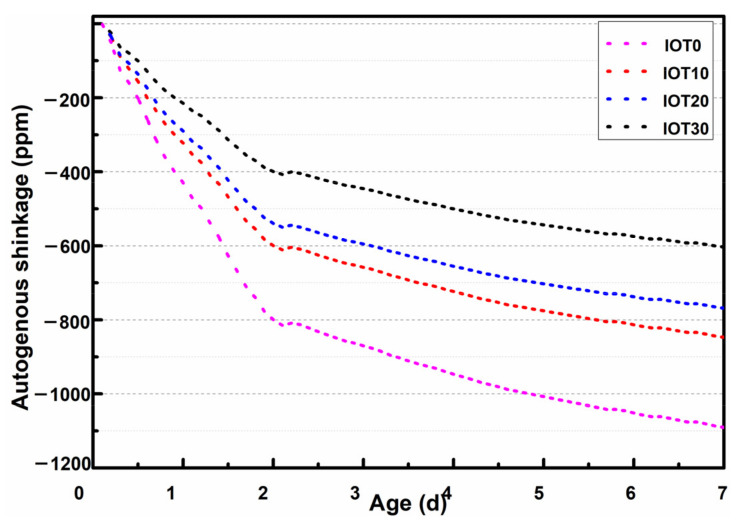
Effect of iron ore tailing on the autogenous shrinkage of designed UHPC.

**Figure 9 materials-14-01829-f009:**
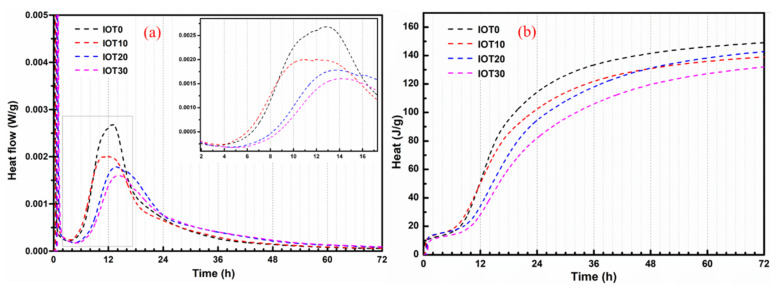
Calorimetry test results of UHPC pastes with different lead-zinc tailings contents: (**a**) Normalized heat flow and (**b**) normalized total heat.

**Figure 10 materials-14-01829-f010:**
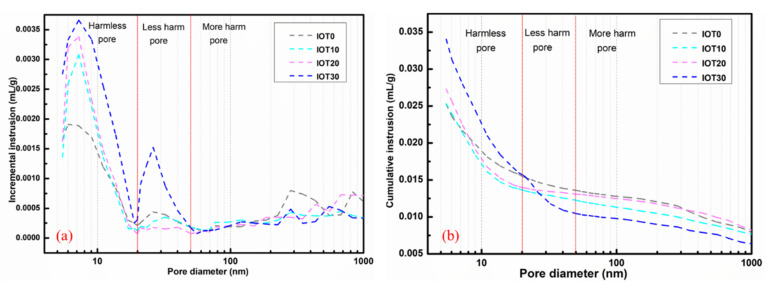
Effect of iron ore tailing on the pore size distribution of designed UHPC at 28 d: (**a**) Incremental intrusion; (**b**) cumulative intrusion.

**Figure 11 materials-14-01829-f011:**
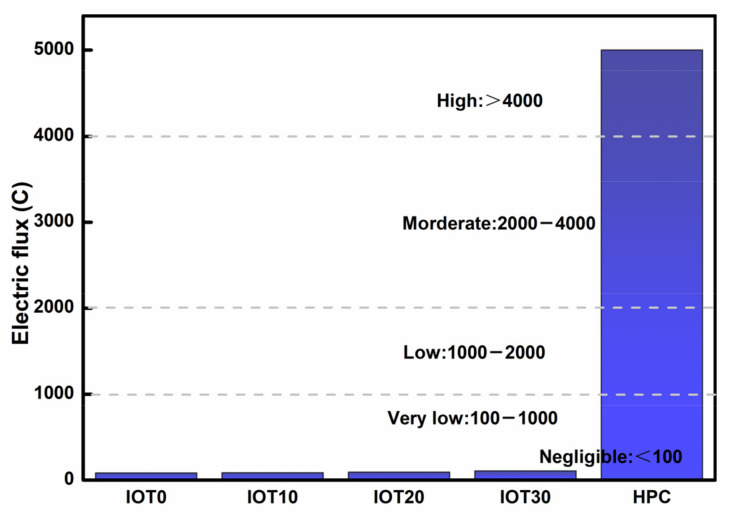
Effect of iron ore tailing on the coulomb electric flux of designed UHPC.

**Figure 12 materials-14-01829-f012:**
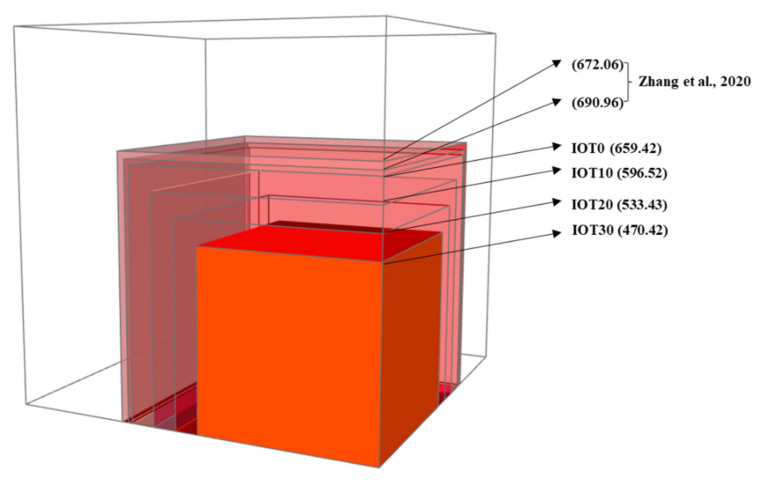
The emission of CO_2_ for different UHPC mixtures.

**Table 1 materials-14-01829-t001:** Chemical compositions of used cementitious materials (wt%).

Compositions	CaO	SiO_2_	Al_2_O_3_	SO_3_	Fe_2_O_3_	Na_2_O	MgO	K_2_O	TiO_2_	P_2_O_5_	LOL
C	63.09	20.14	5.37	2.79	3.14	0.17	1.69	0.73	0.43	0.06	2.06
IOT	13.68	33.26	10.96	10.59	10.11	1.72	6.50	2.31	0.50	0.31	9.60
SF	0.26	94.65	0.15	0.59	0.15	0.23	0.37	0.82	-	0.18	2.39
FA	7.35	46.05	26.58	1.62	7.54	1.35	1.31	2.05	1.51	0.56	3.40

(C: Cement; IOT: Iron ore tailing; SF: Silica fume; FA: Fly ash).

**Table 2 materials-14-01829-t002:** Mix proportion of designed UHPC (kg/m^3^).

No.	C	IOT	FA	SF	S_0–0.6_	S_0.6–1.25_	W	Sp
IOT0	750	0	200	144	770	220	180	36
IOT10	675	75	200	144	770	220	180	36
IOT20	600	150	200	144	770	220	180	36
IOT30	525	225	200	144	770	220	180	36

(C: Cement; IOT: Iron ore tailing; FA: Fly ash; SF: Silica fume; S_0–0.6_: River sand 0–0.6 mm; S_0.6–1.25_: River sand 0.6–1.25 mm; W: Water; PCE: Superplasticizer).

**Table 3 materials-14-01829-t003:** Rheological parameters of the designed UHPC mixture with IOT.

NO.	τ_0_/Pa	m	n	R^2^	μ/(Pa·s)
IOT0	30.00	293.94	0.52	0.9994	74.64
IOT10	28.94	204.53	0.58	0.9983	50.67
IOT20	27.22	175.12	0.65	0.9986	64.25
IOT30	24.89	130.46	0.67	0.9980	61.53

**Table 4 materials-14-01829-t004:** Isothermal calorimetry test results of designed UHPC with IOT.

NO.	End of Introduction Period		Peak of Hydration		Cumulative Heat at 72 h (J/g)
	Q_min_(W/g)	T(Q_min_)(h)	Q_max_(W/g)	T(Q_max_)(h)	
IOT0	2.10 × 10^−4^	3.37	2.68 × 10^−3^	12.57	149.05
IOT10	2.35 × 10^−4^	3.59	2.00 × 10^−3^	10.67	139.01
IOT20	1.83 × 10^−4^	4.55	1.78 × 10^−3^	13.29	142.79
IOT30	1.76 × 10^−4^	4.96	1.60 × 10^−3^	13.69	132.00

## Data Availability

Data is contained within the article. The data presented in this study are available insert article.

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
