# Peer review of "Utilizing Iron Ore Tailing as Cementitious Material for Eco-Friendly Design of Ultra-High Performance Concrete (UHPC)"

_materials, 2021, doi:10.3390/ma14081829_

Round 1
Reviewer 1 Report
This paper presents an extensive experimental study about ultra high performance concrete (UHPC)using iron ore tailing (IOT) as the cementitious materials replacements. The subject is interesting and is compatible with the journal scope. However, some points require some reflection and should be revised. It is recommended that the manuscript be reviewed by someone with fluency in the English language.
In the Introduction chapter, the authors state: It is found that employing iron ore tailings to replace cement can optimize the mechanical properties of concrete [11, 31, 40], and it is pointed out that 40% of cement could be replaced by iron ore tailings when the w/c of high performance concrete is low [40]. As the study carried out replaces up to 30% of cement, it is unclear what new the paper intends to explore.
So, it is recommended to underline what the existing and cited literature did not clarify and which is intended to be clarified with this study.
- Materials and methods
In 2.2. Exprimental methodogy, replace by Experimental methodology
What are the units in table 2 ?
2.2.2 Workability
If the test applied to evaluate the workability is normalized, please cite the standard norm.
The rheological profile shown in Fig. 5 looks strange. What do the dotted and thick lines mean?
- Results and discussion
Fig 7, please fix share by shear.
Review the text after table 3.
« Hence, it could be summarized from the slump flow and rheology tests that it is logical to
adopt IOT as cementitious materials during the preparing of UHPC, as IOT can optimize the
workability of UHPC when moderate IOT is utilized. » This conclusion of subchapter 3.2 is also strange. What the authors could said with logical and moderate?
In subchapter 3.3 the authors said: … « the pozzolanic effect of the fine IOT particles will deplete the
superfluous Ca(OH)2 and generate more C-S-H gel. » On what basis do the authors consider that there is a pozzolanic effect at young ages?
In sub-chapters 3.6 and 3.7, the authors mention the possibility of some pozzolanic reaction. Again, one wonders on, what basis is any pozzolanic reaction justified in the early ages?
Although the mechanical results of the mixtures studied here are high, in the case of mortars, one wonders: Would these magnitudes of values could be obtained if the compressive strength tests were carried out on specimens of dimensions common to concrete?
Conclusions
revise according to the changes inserted in the previous chapters, if applicable.
Author Response
Response to Reviewer 1 Comments
Piont 1: This paper presents an extensive experimental study about ultra high performance concrete (UHPC)using iron ore tailing (IOT) as the cementitious materials replacements. The subject is interesting and is compatible with the journal scope. However, some points require some reflection and should be revised. It is recommended that the manuscript be reviewed by someone with fluency in the English language.
Response 1: Thank you for your useful comments and valuable suggestions on our manuscript. Detailed modification has been made according to the comments in revised manuscript. The English language has been reviewed carefully in the revised manuscript.
Piont 2: In the Introduction chapter, the authors state: It is found that employing iron ore tailings to replace cement can optimize the mechanical properties of concrete [11, 31, 40], and it is pointed out that 40% of cement could be replaced by iron ore tailings when the w/c of high performance concrete is low [40]. As the study carried out replaces up to 30% of cement, it is unclear what new the paper intends to explore. So, it is recommended to underline what the existing and cited literature did not clarify and which is intended to be clarified with this study.
Response 2: Thank you for your constructive comment. We are sorry for the confusion. The purpose and significance of this study are underlined. It is pointed out in reference [40] that it is feasible to add IOT into ordinary concrete. But there are significant differences in the design method and the composition of cementitious materials between UHPC and ordinary concrete. Therefore, the feasibility of preparing UHPC by utilizing high volume of IOT to replace cement is still need demonstration.
Piont 3:
2.Materials and methods
In 2.2. Exprimental methodogy, replace by Experimental methodology
What are the units in table 2?
Response 3: Thank you for your carefully reading. The review is right. The “Exprimental methodogy” in 2.2 has been changed into “Experimental methodology”. The units in table 2 has been added.
Piont 4:
2.2.2 Workability
If the test applied to evaluate the workability is normalized, please cite the standard norm.
Response 4: Thank you for your valuable suggestions. The review is right. The workability test is carried out according to the Standard of BS-EN-1015-3, and it has been added in the revised manuscript.
Piont 5: The rheological profile shown in Fig. 5 looks strange. What do the dotted and thick lines mean?
Response 5: Thank you for your useful comments and valuable suggestions on our manuscript. We are sorry for the confusion. The rheological profile shown in Fig. 5 has been modified. The original Fig. 5 is a dot chart, the blue line indicates the change of shear rate with time, while the gray dotted line is just for visualization.
Piont 6:
- Results and discussion
Fig 7, please fix share by shear.
Review the text after table 3.
Response 6: The authors thank this reviewer for the carefully reading, the reviewer is right. The share in Fig. 7 has been fixed by shear. The text after table 3 has been simplified and reorganized.
Piont 7: “Hence, it could be summarized from the slump flow and rheology tests that it is logical to adopt IOT as cementitious materials during the preparing of UHPC, as IOT can optimize the workability of UHPC when moderate IOT is utilized.” This conclusion of subchapter 3.2 is also strange. What the authors could said with logical and moderate?
Response 7: The authors thank this reviewer for the valuable comments, the reviewer is right. The conclusion of subchapter 3.2 has been revised and rewritten. “Hence, it could be summarized from the slump flow and rheology tests that the addition of IOT is beneficial to reduce the viscosity and optimize the workability of UHPC.”
Piont 8: In subchapter 3.3 the authors said:“the pozzolanic effect of the fine IOT particles will deplete the superfluous Ca(OH)2 and generate more C-S-H gel.” On what basis do the authors consider that there is a pozzolanic effect at young ages?
Response 8: The authors thank this reviewer for the useful and valuable comments, the reviewer is right. The activity of IOT is relatively low in the early age [1]. But he IOT can be activated by mechanical grinding, and the IOT used in this study is ground power, and its activity is higher. Moreover, the increase of compressive strength from 7 d to 28 d for IOT0, IOT10, IOT20 and IOT30 are 25.7%, 45.1%, 46.5% and 55.2%, respectively. The increment of UHPC with IOT is more significant. Based on the above experimental phenomenon,it is suggested that the improvements at 28 d are due to the filling effect and pozzolanic effect of IOT [2-3].
- Han, F. H.; Li, L.; Song, S. M.; Liu, J. H., Early-age hydration characteristics of composite binder containing iron tailing powder. Powder Technol 2017, 315, 322-331.
- Cheng, Y. H.; Huang, F.; Li, W. C.; Liu, R.; Li, G. L.; Wei, J. M., Test research on the effects of mechanochemically activated iron tailings on the compressive strength of concrete. Constr Build Mater 2016, 118, 164-170.
- Ma, B. G.; Cai, L. X.; Li, X. G.; Jian, S. W., Utilization of iron tailings as substitute in autoclaved aerated concrete: physico-mechanical and microstructure of hydration products. J Clean Prod 2016, 127, 162-171.
Piont 9: In subchapters 3.6 and 3.7, the authors mention the possibility of some pozzolanic reaction. Again, one wonders on, what basis is any pozzolanic reaction justified in the early ages?
Response 9: The authors thank this reviewer for the serious comment. The MIP test in subchapter 3.6 and coulomb electric flux test in subchapter 3.7 are conducted at age of 28 d. The optimization of pore size distribution at 28 d are related to the filling and nucleation effect of fine IOT practices. Besides, the chemical activity (pozzolanic reaction) of ground IOT also has some beneficial effects. The pozzolanic reaction of ground IOT has been confirmed by TG-DTA analysis and EDS test in reference [2].
Piont 10: Although the mechanical results of the mixtures studied here are high, in the case of mortars, one wonders: Would these magnitudes of values could be obtained if the compressive strength tests were carried out on specimens of dimensions common to concrete?
Response 10: The authors thank this reviewer for the valuable comments. This involves the size effect of concrete. The measured compressive strength of UHPC samples with the size of 100 mm × 100 mm × 100 mm is lower than that of mortar samples, but the reduction is less than 20% according our test results. And the compressive strength for UHPC sample is still higher than 100 MPa when the tests are carried out on specimens of dimensions common to concrete. The size effect is explained in subchapter 3.3.
Piont 11: Conclusions revise according to the changes inserted in the previous chapters, if applicable.
Response 11: The authors thank this reviewer for the useful comments. The conclusion has been revised according to the changes inserted in the previous chapters.

Reviewer 2 Report
I recommend the paper " Utilizing iron ore tailing as cementitious material for eco-friendly design of ultra high performance concrete (UHPC)” for publication. Nevertheless, the paper required some corrections and additions.
Detailed report and comments:
- Page 2. “It has beendemonstrated thatiron ore tailing can beappliedto replacefine aggregates in the production of concrete materials (Han et al., 2017; Liu et al., 2012).” The format citations should be corrected.
- Page 3. “Portland cement (P â…¡525 according to Chinese standard GB/T175-2007)…”. The reference (proper citation) to Standard should be added.
- Page 3. Table 2. The units are missed.
- Page 6. “The samples with the measurementof 40 mm ×40 mm ×160 mm…” The citation/reference to the standard should be given. The samples were prepared according to what standard?
- Page 6. “and its pressures ranging from 0.5 psi to 55,000 psi.” The described ranging should be given in SI units (MPa).
- Fig. 7 and Table 3. Move the table and drawing behind the first reference.
- Page 8. “The rheological parameters are calculated and displayed in table 2.” The proper number of Table is 3.
- Page 11. The listed in-text Table numbers should be verified and corrected.
- Figures should be supplemented with grids for proper visualizations.
- The conclusions should be supplemented to indicated the eco-friendly character of mixed concrete design.
Author Response
Response to Reviewer 2 Comments
Piont 1: Page 2. “It has beendemonstrated thatiron ore tailing can beappliedto replacefine aggregates in the production of concrete materials (Han et al., 2017; Liu et al., 2012).” The format citations should be corrected.
Response 1: Thank you for your carefully reading. We are sorry for the confusion. The format citations have been corrected. “It has been demonstrated that iron ore tailing can be applied to replace fine aggregates in the production of concrete materials [11, 31].”
Piont 2: Page 3. “Portland cement (P â…¡525 according to Chinese standard GB/T175-2007)…”. The reference (proper citation) to Standard should be added.
Response 2: Thank you for your useful comments and valuable suggestions on our manuscript. The reference of Standard GB/T 175-2007 has been added.
Piont 3: Page 3. Table 2. The units are missed.
Response 3: The authors thank this reviewer for the careful reading. We are sorry for the mistake. The unites in Table 2 has been added.
Piont 4: Page 6. “The samples with the measurement of 40 mm ×40 mm ×160 mm…” The citation/reference to the standard should be given. The samples were prepared according to what standard?
Response 4: Thank you for your constructive comment, the reviewer is right. The standard BS-EN-196-1 for the preparation and testing method of samples has been added.
Piont 5: Page 6. “and its pressures ranging from 0.5 psi to 55,000 psi.” The described ranging should be given in SI units (MPa).
Response 5: Thank you for your carefully reading, the reviewer is right. The unit of psi has been converted to MPa. “and its pressures ranging from 0.0034 MPa to 379.225 MPa.”
Piont 6: Fig. 7 and Table 3. Move the table and drawing behind the first reference.
Response 6: The authors thank this reviewer for the useful suggestion, the reviewer is right. Fig. 7 and Table 3 have been moved to the end of the first reference.
Piont 7: “The rheological parameters are calculated and displayed in table 2.” The proper number of Table is 3.
Response 7: The authors thank this reviewer for the useful suggestion, the reviewer is right. The mistake has been corrected.
Piont 8: Page 11. The listed in-text Table numbers should be verified and corrected.
Response 8: Thank you for your useful comments and valuable suggestions on our manuscript. The listed in-text Table numbers have been verified and corrected carefully.
Piont 9: Figures should be supplemented with grids for proper visualizations.
Response 9: The authors thank this reviewer for the useful suggestion, the reviewer is right. The grids for Fig. 6, Fig. 8, Fig. 9, Fig. 10 have been supplemented.
Piont 10: The conclusions should be supplemented to indicated the eco-friendly character of mixed concrete design.
Response 10: The authors thank this reviewer for the useful suggestion, the reviewer is right. The eco-friendly character of mixed concrete design in conclusions has been supplemented.

Round 2
Reviewer 1 Report
After reading the revised version by the authors, I consider that the manuscript has been substantially improved with the clarifications made.
Author Response

(The authors gave the same response as above.)
